# Enhanced Two-Photon Fluorescence and Fluorescence Imaging of Novel Probe for Calcium Ion by Self-Assembly with Conjugated Polymer

**DOI:** 10.3390/polym11101643

**Published:** 2019-10-10

**Authors:** Yue-liang Zhai, Qiu-bo Wang, Hao Yu, Xiao-yuan Ji, Xian Zhang

**Affiliations:** 1School of Materials Science and Engineering, Qilu University of Technology (Shandong Academy of Sciences), Jinan 250353, China; zhaiyuelianger@163.com (Y.-l.Z.); wangqbqlu18@163.com (Q.-b.W.); yh15215425480@163.com (H.Y.); jixy0430@163.com (X.-y.J.); 2Shandong Provincial Key Laboratory of Processing and Testing Technology of Glass and Functional Ceramics; Key Laboratory of Amorphous and Polycrystalline Materials; Qilu University of Technology, Jinan 250353, China

**Keywords:** fluorescence probe, Ca^2+^, conjugated polymer, self-assembly, two-photon fluorescence imaging

## Abstract

The calcium ion (Ca^2+^) isa highly versatile intracellular signal messenger regulating many different cellular functions. It is important to design probes with good fluorescence and two-photon (TP) active cross-sections (Φ*δ*) to explore the concentration distribution of Ca^2+^. In this manuscript, a novel TP fluorescence calcium probe (BAPTAVP) with positive charges, based on the classical Ca^2+^ indicator of BAPTA (1,2-bis(2-aminophenoxy)-ethane-*N,N,N’,N’*-tetra acetic acid), and a conjugated polymer (PCBMB) with negative charges were designed and synthesized. The results from transmission electron microscopy (TEM), atomic force microscopy (AFM), dynamic light scattering (DLS), and the zeta potential (ZP) showed that nanoparticles were obtained by the self-assembly of PCBMB and BAPTAVP. Moreover, the fluorescence properties of BAPTAVP were effectively improved by fluorescence resonance energy transfer (FRET) with PCBMB and attenuating the intramolecular charge transfer (ICT) after the addition of Ca^2+^. The quantum yield and Φ*δ* of PCBMB-BAPTAVP increased by about four and six times in comparison to those of BAPTAVP, respectively. The TP fluorescence imaging experiments indicated that the PCBMB-BAPTAVP system could effectively detect Ca^2+^ in living cells with high sensitivity.

## 1. Introduction

Recently, research about selective and sensitive fluorescent indicators for metal ions has attracted significant attention [1,2,3], because metal ions play indispensable roles in a variety of fundamental physiological processes in organisms. The calcium ion, Ca^2+^, is a highly versatile intracellular signal messenger regulating many different cellular functions [4,5], such as sensory transduction, exocytosis, muscle contraction, and enzyme activity [6,7]. The change inCa^2+^concentrations and the translocation of Ca^2+^ across the plasma membrane have often been used to explore these functions [8]. At present, two-photon microscopy (TPM) combined with suitable probes exhibits better localized excitation, much less photon damage and photobleaching [9], a small absorption coefficient of light in tissue, and lower tissue autofluorescence [10], and is often used to study Ca^2+^ in living cells. Although a few two-photon (TP) fluorescence probes such as ACaLN have been developed [11], and their properties are far superior to previous one-photon (OP) fluorescence ones, such as Mag-Fura-2 and Calcium Green C_18_ applied in TPM, the TP active cross-sections (Φ*δ*) of these probes are still not very ideal [12]. Thus, it is important to develop new suitable probes with larger Φ*δ* using the simple method. 

As we all know, conjugated polymers (CPs), as an important type of TP material, exhibit good fluorescent properties. They usually have the advantages of a large Stokes shift, high quantum yield (Φ), excellent optical stability, and reasonable application prospects in the field of fluorescence imaging [13,14]. However, their targeting has been questioned, limiting further application [15,16,17], whereas many small organic molecules have exhibited a high performance at recognizing the targets [18,19]. Thus, we think it is possible to improve the TP properties of dyes by combining small molecules with CPs. At present, some novel two-photon fluorescence sensing platforms based on fluorescence resonance energy transfer (FRET) between the dyes and CPs have been designed to achieve better fluorescence properties [20,21]. Zhang et al. obtained a novel fluorescent probe by FRET between gold and conjugated polymer nanoparticles. The probe was highly sensitive and selective to melamine and has been successfully applied to milk powder [22]. Sha et al. designed a spirolactam rhodamine-linked adamantine as a ratiometric sensor for Hg^2+^. The fluorescence was effectively enhanced by FRET on the polymer brush-functionalized mesoporous silica nanoparticles with CPs [23]. However, few studies have detected Ca^2+^ based on the polymers by two-photon excitation. It may be a strategy to improve the TP properties of dyes by combining small molecules with CPs.

Herein, a new TP probe (BAPTAVP) with positive charges based on the traditional indicator of Ca^2+^ with high selectivity and sensitivity was synthesized and characterized. Nanoparticles were formed by self-assembly with the conjugated polymer of a carbazole derivative (PCBMB) with negative charges and BAPTAVP with positive charges. Moreover, TP fluorescent properties of BAPTAVP were effectively improved by FRET with PCBMB and attenuating the intramolecular charge transfer (ICT) after the addition of Ca^2+^. The key diagram is displayed in Scheme 1. Furthermore, TPM for the detection of Ca^2+^ in living cells was finished successfully. It may provide a new method to improve the TP properties of some traditional fluorescence probes and further widen their applications.

## 2. Experimental

### 2.1. Chemicals and Instruments

All chemicals were obtained from commercial suppliers and used without further purification. Ultrapure water was used in all experiments. A solution of Ca^2+^ was prepared from their nitrate salts.

^1^H NMR spectra were recorded on a Bruker AvanceII 400 spectrometer using TMS as an internal standard. The element analyses were performed on a Perkin 2400 (II) autoanalyzer. Infrared spectra were recorded on a Nicolet NEXUS 670 FT-IR spectrometer using a liquid-nitrogen-cooled detector. UV-vis absorption spectra were recorded on a Varian Cary 50 spectrophotometer using a quartz cuvette with a 1 cm path length. The pH values were measured with a Mettler-Toledo FE20 pH meter. One-photon fluorescent spectra were recorded with a HITRCHIF-7000 fluorescence spectrometer. Two-photon fluorescent spectra were recorded on an OOIBASE32 spectrophotometer. The pump laser beam was from a mode-locked Ti:Sapphire laser system with a pulse duration of 200 fs, a repetition rate of 76 MHz, and a wavelength range of 700–1000 nm (Coherent Mira900-D). Transmission electron microscopy (TEM) images were recorded through a JEM-2100 electron microscope (JEOL Ltd., Musashino, Japan). Atomic force microscopy (AFM) was performed on a Multimode8. Dynamic light scattering (DLS) and zeta potential (ZP) data were obtained on a Malvern Zetasizer Nano ZS analyzer (Malvern, Worcestershire, UK). Fluorescence images of SiHa cells were obtained using a Zeiss Ism710 laser confocal microscope.

### 2.2. Synthesis

For the synthesis details of 5-formyl-5′-methyl-BAPTA ethyl ester (**1**) (BAPTA: 1,2-bis(2-aminophenoxy)-ethane-*N,N,N’,N’*-tetra acetic acid)and 4-methyl-*N*-methyl pyridine iodized salt (**2**), see the relevant literature [24,25].

We obtained 1,4-bischloromethyl-2, 5-dimethoxybenzene (**3**), and [(2,5-dimethoxy-1,4-phenylene)]bis-(methylene)]bistriphenylphosphonium dichloride (**4**) as described in the literature [26]. The synthesis process of *N*-bromoethyl-carbazole (**5**) was referred in the literature [27].

The synthetic routes of BAPTAVP and PCBMB are shown in Scheme 2 and Scheme 3, respectively.

#### 2.2.1. Procedure for the Synthesis of 5-(4-vinyl-N-methyl pyridine iodized salt)-5′-Methyl-BAPTA Ethyl Ester (BAPTAVP)

Aldehyde derivative 1 (1 g, 1.587 mmol) and iodized salt 2 (0.39 g, 1.66 mmol) were dissolved in ethyl alcohol (30 mL). After adding 10 drops of piperidine, the mixture was slowly heated to 80 °C and stirred for 10 h, followed by cooling to room temperature. After filtering and washing with water and methanol, the crude product was obtained and then recrystallized to give yellow crystals with a yield of 47.3%. ^1^H NMR (CDCl_3_, 400 MHz, TMS) δ (ppm): 8.54 (d, *J* = 7.2 Hz, 2 H), 8.03 (d, *J* = 6.8 Hz, 2 H), 7.25 (m, 4 H), 6.76 (m, 4 H), 4.21 (m, 20 H), 2.28 (s, 3 H), 1.94 (s, 3 H), 1.18 (t, *J* = 22 Hz, 12 H). IR (KBr) ν/cm^−1^: 1632 cm^−1^ (C=C); 1253 cm^−1^ (C–N); 1591 cm^−1^ (C=N); 2928 cm^−1^, 2994 cm^−1^ (–CH_3_); 1738 cm^−1^ (C=O); 1009 cm^−1^ (C–O). Element analysis calcd (%) for C_39_H_50_N_3_O_10_I: C 55.25, H 5.90, N 4.96; found: C 55.38, H 5.88, N 4.98.

#### 2.2.2. Procedure for the Synthesis of 3,6-bis-(methyl ketone)-N-bromoethyl-carbazole (**6**)

In a three-necked bottle, anhydrous aluminum trichloride (0.813 g, 7.10 mmol) was dissolved in 1,2-dichloroethane (13.5 mL) and then cooled to 5 °C. Acetyl chloride (0.96 g, 12.2 mmol) was added to the mixture in two minutes. Compound 5 (1.4 g, 5.30 mmol) dissolved in 1,2-dichloroethane was dropwise added into the reactive solution and stirred for 20 min. Then, the mixture was warmed up to 35 °C and stirred for another 2.5 h. The solvent was removed under reduced pressure. The mixture was neutralized with diluted hydrochloric acid (15 mL) and filtered to afford pale yellow powders, and then recrystallized from ethanol. ^1^H NMR (CDC1_3_, 400 MHz, TMS) δ (ppm): 8.80(s, 2 H), 8.20(d, *J* = 10.4 Hz, 2 H), 7.51(d, *J* = 8.4 Hz, 2 H), 4.79(t, *J* = 14.4 Hz, 2 H), 3.75(t, *J* = 14 Hz, 2 H), 2.76(s, 6 H).

#### 2.2.3. Procedure for the Synthesis of the Conjugated Polymer (PCBMB)

A mixture including compound 4 (0.758 g, 0.997 mmol), compound 6 (0.36 g, 1.05 mmol), and chloroform (10 mL) was added to a three-necked bottle and stirred for 30 min at room temperature(r.t.) under nitrogen atmosphere. Potassium tert-butoxide (1 g, 8.93 mmol) dissolved in ethanol was dropwise added to the mixture and stirred for 2 days at r.t., and the solvent was then removed under reduced pressure. The residue was dissolved in dichloromethane and stirred for 10 min, then filtered. The yellow powers were obtained after washing with methyl alcohol three times with a yield of 30.4%. ^1^H NMR(CDC1_3_, 400 MHz, TMS) δ (ppm): 8.78(s, 2 H), 8.18(d, *J* = 10.4 Hz, 2 H), 7.69(d, *J* = 8.4 Hz, 2 H), 7.28(s, 2 H), 6.69(m, 1 H), 5.70(d, *J* = 16.8 Hz, 1 H), 5.45(d, *J* = 10 Hz, 1 H), 3.70(m, 2 H), 2.76(s, 6 H). IR (KBr) ν/cm^−1^: 3005 cm^−1^ (C–H); 1267 cm^−1^ (C–N); 2949 cm^−1^, 3001 cm^−1^ (CH_3_); 1686 cm^−1^ (C=O); 1675 cm^−1^ (C=C); 1044 cm^−1^ (C–O); 626 cm^−1^ (C–Br). M_w_/M_n_ = 1.326, M_z_/M_n_ = 1.809; M_n_ = 5127; M_w_ = 5820; M_z_ = 6848 (GPC, polystyrene calibration).

### 2.3. Preparation of Solutions and Spectral Measure

The stock solutions of Ca^2+^ in the ultrapure water and BAPTAVP were prepared in tris-HCl buffer solution (10 mmol/L, KCl 100 mmol/L, pH 7.2) with a concentration of 2 × 10^−2^ mol/L. The stock solution of PCBMB (1 × 10^−3^ mol/L) was prepared in THF. Test sample solutions were diluted to the appropriate concentrations using tris-HCl buffer solution, and the spectra were then measured after adding acetylcholin esterase and placed for 20 min. The OP and TP fluorescence spectra were excited at 264 nm (for PCBMB), 420 nm (for BAPTAVP), and 800 nm, respectively. The ^1^H NMR of three samples including pure BAPTAVP, BAPTAVP-added acetylcholin esterase, BAPTAVP-added acetylcholin esterase, and Ca^2+^wasmeasured in CDCl_3_. 

### 2.4. General Processes for Cell Culture and Fluorescence Imaging

Siha cells were grown in H-DMEM (Dulbecco’s Modified Eagle’s Medium, High Glucose) supplemented with 10% FBS (fetal bovine serum) in a 5% CO_2_ incubator at 37 °C. Cells (1 × 10^5^ / mL) were placed on glass coverslips and allowed to adhere for 24 h. The living SiHa cells were incubated with 5 μmol/L solution of BAPTAVP, BAPTAVP-Ca^2+^, PCBMB-BAPTAVP, and PCBMB-BAPTAVP-Ca^2+^ for 1h at 37 °C, respectively, and were then washed with PBS three times to image. Two-photon fluorescence imaging was observed under a Zeiss Ism710 confocal microscope at 800 nm.

## 3. Results and Discussion

### 3.1. Design of BAPTAVP and PCBMB

The probe was designed on the basis of BAPTA (o,o’-bis(2-aminophenyl) ethyleneglycol-N,N,N’,N’-tetra acetic acid), which is a well-known Ca^2+^ indicator that has been widely applied as OP fluorescence probes [28]. By introducing four ester groups (-Et), the penetrability of the probe into cells can be considerably improved [29,30]. The pyridine ring as an excellent electron acceptor was introduced into the structure. Biphenyl ethylene was chosen as the conjugated bridge owing to its good planarity. Meanwhile, in order to improve biocompatibility, the compound was converted to pyridine salt. The design of PCBMB was based on carbazole derivatives owing to its π-conjugated backbone. The delocalized system was further enlarged by introducing the benzene ring and conjugated double-bond by the Wittig reaction. The fluorescent intensity was further strengthened by connecting a strong electron donor of -OCH_3_.

### 3.2. Linear and Nonlinear Spectra of BAPTAVPwith Ca^2+^Titration

Different volumes of Ca^2+^ were added to the solution of BAPTAVP in tris-HCl buffer solution (10 mmol/L, KCl 100 mmol/L, pH 7.2) with a concentration of 1 × 10^−5^ mol/L, and the mole concentration ratios of Ca^2+^ to BAPTAVP (Ca^2+^:B) were 0, 0.2, 0.5, 1.0, 2.0, 3.0, 6.0, 10.0, 15.0, and 20.0. The absorption and OP fluorescence spectra are shown in Figure 1a,b, respectively. It can be observed that the changing trend of absorbance is almost coincident with that of the fluorescent intensity (upper right corner in Figure 1a,b). The absorbance and fluorescence obviously increase with the addition of Ca^2+^ before the ratio of Ca^2+^/probe = 1, and then begin to decline and level off at the ratio of 8.0, which indicates that a steady complex of Ca^2+^/probe may have formed at the ratio of 1.0. Meanwhile, other coordinations of Ca^2+^ to BAPTAVP, such as a 2:1 or 3:1 complex (Ca/B), may appear to induce a change in fluorescence. A similar change trend was found in other research [31]. The fluorescent intensity of BAPTAVP exhibits a good linear relationship (y = 1.43x + 46.7, R^2^ = 0.991) with the concentrations of Ca^2+^ in the range from 1 × 10^−7^ to 1 × 10^−5^ mol/L in Figure 1c. The limit of detection (LOD) of BAPTAVP for OP fluorescence was estimated to be 9.6 × 10^−8^ mol/L, which was obtained by the equation LOD = 3S_b_/K [32] (S_b_ represents the standard deviation and K is the slope between fluorescence intensity versus Ca^2+^ concentration). This value is superior to that reported in the literature in the detection of endogenous Ca^2+^ [33].

TP fluorescent spectra were investigated upon adding different volumes of Ca^2+^ to tris-HCl buffer solution with a concentration of 8 × 10^−4^ mol/L at 800 nm. The mole ratios of Ca^2+^/probe were 0, 0.2, 0.5, 1.0, 2.0, 3.0, 6.0, and 10.0. The experimental results are displayed in Figure 1d. Compared to the original compound, no obvious red-shifts or blue-shifts were found with the increase in Ca^2+^ concentration. The changing trend of TP fluorescent intensity is similar to that of OP fluorescence. However, the fluorescent maximum point appeared at the ratio of Ca^2+^/probe = 0.5, which is different from that of OP excitation. This may be attributed to the higher concentration (8 × 10^−4^ mol/L) of the dye under TP excitation, and this phenomenon was found in our previous reports [18,34].

### 3.3. Dissociation Constant of BAPTAVP/Ca^2+^Complex

The dissociation constant (*Kd*), which represents the connection between Ca^2+^ and the dye, was calculated. The *Kd* was calculated according to the following equation [28]:(1)$$\log\;\left[{\mathrm{Ca}}^{2+}\right]=\log k_{d}+\log\left[\left(F-F_{\max}\right)/\left(F_{\min}-F\right)\right]$$
where *F* is the fluorescence intensity, *F_max_* is the maximum fluorescence intensity, *F_min_* is the fluorescence intensity in the absence of Ca^2+^, and [Ca^2+^] is the free calciumion concentration. The *Kd* value can be obtained from the fitting curve (see Figure 2a).

The value is 0.982 μmol/L, which is larger than those of some traditional Ca^2+^ probes, such as Fluo-3 (*Kd* = 0.39 μmol/L) [35]. This indicates that the BAPTAVP has a lower-affinity (higher *Kd* value) fluorescence probe for Ca^2+^. Although the fluorescence indicators with high affinity (low *Kd* value) exhibit stronger fluorescence, the fluorescence change would be delayed because of the low dissociation. The fluorescence of the indicators easily reached saturation under the lower concentrations of Ca^2+^, which would cause a larger deviation to analyze intracellular Ca^2+^. As a result, the fluorescence probes for intracellular Ca^2+^ with a lower affinity (higher *Kd* value) have been widely applied in fluorescence microscopy [33,36].

### 3.4. The Effects of Different Volumes of PCBMBon the Fluorescence of BAPTAVP

In Figure 2b, there are obvious overlaps between the fluorescence emission spectrum of PCBMB and UV absorption spectrum of BAPTAVP. According to the related literature [37,38], FRET may occur between the PCBMB (donor) and BAPTAVP (acceptor). To figure out the influence of adding PCBMB on the fluorescence of BAPTAVP, different volumes of PABMB were added to the solution of BAPTAVP in the tris-HCl buffer solution (10 mmol/L, KCl 100 mmol/L, pH 7.2) with a concentration of 1 × 10^−5^ mol/L, and the mole ratios of PCBMB to BAPTAVP (P:B) were 0, 0.2, 0.5, 1.0, 2.0, 3.0, 6.0, 10.0, 15.0, and 20.0. The UV-vis absorption spectra and OP fluorescence spectra are shown in Figure 2c,d, respectively. It can be seen from Figure 2c that the maximum absorption peak of BAPTAVP is at 420 nm, and that of PCBMB is at 325 nm. A new absorption peak at 340 nm appears when adding PCBMB into the solution of BAPTAVP. With the addition of PCBMB, the new absorption at 340 nm continuously increases, while the absorption of BAPTAVP at 420 nm gradually decreases and disappears. Similar phenomena are found in Figure 2d. A new fluorescence emission peak was obtained in comparison to those of BAPTAVP and PCBMB. The sharp bands at 528 nm are mainly the second harmonics of the beam source. Their amplitudes are enhanced with the increased concentrations of PCBMB, which have been reported in other studies [32,39]. Moreover, the fluorescent intensity increased dramatically with the addition of PCBMB before the ratio of polymer/probe = 1, and then levelled off. It was inferred that the self-assembly process between PCBMB with negative chargesand BAPTAVP with positive charges was carried out by electrostatic interaction. Moreover, the charge transfer (CT) took place from electron donor (PCBMB) to electron acceptor (BAPTAVP), and the new blue-shift absorption peak and fluorescence emission peak appeared in comparison to those of BAPTAVP. A similar CT process was reported in the literature [40,41]. In addition, the distance between BAPTAVP and PCBMB will shorten after self-assembly, and extensive overlaps between the fluorescence emission spectrum of PCBMB and UV absorption spectrum of BAPTAVP in Figure 2b present an opportunity for FRET to greatly enhance the fluorescence of BAPTAVP. Furthermore, Figure 2d indicates that a steady complex of polymer-probe may form at the ratio of 1.0. This implies that the addition volume of PCBMB to achieve the ratio of BAPTAVP:PCBMB = 1:1 was optimal.

### 3.5. The Enhanced Fluorescence of Probe-Polymer System Detection of Ca^2+^

OP fluorescent emission spectra of PCBMB, BAPTAVP, BAPTAVP-Ca^2+^ (1:1), PCBMB-BAPTAVP (1:1), and PCBMB-BAPTAVP-Ca^2+^ (1:1:1) are described in Figure 3a. The TP fluorescence of PCBMB was not found at 800 nm, which can be attributed to the poor absorbance at 400 nm (see Figure 2c). TP fluorescent spectra of the other four materials are shown in Figure 3b. Figure 3 shows that a new fluorescent emission band for PCBMB-BAPTAVP appeared in comparison to those of the original BAPTAVP and PCBMB, in both OP excitation and TP excitation. The experimental photophysical data are summarized in Table 1. The fluorescent quantum yields (Φ*_s_*) and two-photon cross-sections (Φ*δ*) of the samples were measured by using coumarin307 in methanol (Φ = 0.56, Φ*δ* = 27.7 GM at 800 nm) standard as the reference [42,43].

The OP fluorescent peak of PCBMB-BAPTAVP was blue-shifted by106 nm and red-shifted by 51 nm in comparison to those of original BAPTAVP and PCBMB, respectively, while its TP fluorescent peak was only blue-shifted by37 nm in comparison to that of BAPTAVP. The reason may be attributed to the higher concentration (1 × 10^−4^ mol/L) of the system solution to achievereabsorption, and this influence was more remarkable for the system of PCBMB-BAPTAVP. The effect of concentration was investigated and is shown in Appendix A. The concentrations of the polymer-probe (PB:PCBMB:BAPTAVP = 1:1) were 5 × 10^−4^ mol/L (PB-1), 1 × 10^−4^ mol/L (PB-2), 5 × 10^−5^ mol/L (PB-3), and 1 × 10^−5^ mol/L (PB-4). The blue-shift degree of TP fluorescene peaks (PB-4, 475 nm; PB-3, 537 nm; PB-2, 569 nm; PB-1, 587 nm)will decline with increased concentrations of PCBMB-BAPTAVP versus the fluorescene emission peak of BAPTAVP.Φ*δ* of BAPTAVP was slightly larger than that of coumarin 307 (15 GM) [43], and the fluorescent quantum yield was dramatically smaller than that of coumarin 307 (0.56), which indicates that the fluorescent properties of BAPTAVP were not very ideal. Considering CPs as TP fluorescent materials have a larger molar absorption coefficient, high fluorescence quantum yields, and outstanding TP cross-sections [44,45], we expected that the performance of BAPTAVP could be improved by doping a conjugate polymer with strong fluorescence. We can see that the fluorescence of BAPTAVP was greatly increased by combining PCBMB. In Table 1, the quantum yield and Φ*δ* of PCBMB-BAPTAVP were increased by approximately 4 times (from 0.098 to 0.39) and 5.5 times (from 16.2 GM to 89 GM) in comparison to those of BAPTAVP, respectively. After adding Ca^2+^, the two values were 0.65 and 169, respectively, which indicates that the system of PCBMB-BAPTAVP was more sensitive to the detection of Ca^2+^.

### 3.6. Fluorescence Enhanced Mechanism

All those changes suggest that the fluorescence and sensitivity of the probe for Ca^2+^ can be effectively improved by combining PCBMB. The possible interaction mechanism was, furthermore, explored by TEM, DLS, and AFM, as shown in Figure 4 and Table 1. The average particle diameters of PCBMB-BAPTAVP and PCBMB-BAPTAVP-Ca^2+^ were about 25 and 58 nm in Figure 4a,b, respectively, while that of pure BAPTAVP was not obtained, and the small molecule of BAPTAVP may be the source. In addition, TEM images in Figure 4e,f show that obvious sphere particles were found after adding the polymer of PCBMB. With the further addition of Ca^2+^, the obvious aggregation of PCBMB-BAPTAVP particles will appear. This is coincident with the change in the particle diameters (from 25 nm of PCBMB-BAPTAVP to 58 nm of PCBMB-BAPTAVP-Ca^2+^). The AFM images present clearer evidence in Figure 4g–l. The nanoparticle sizes of PCBMB-BAPTAVP are more uniform. The obvious aggregation was found after the addition of Ca^2+^. Furthermore, the ZPs of different systems including PCBMB, BAPTAVP, BAPTAVP-Ca^2+^, PCBMB-BAPTAVP, and PCBMB-BAPTAVP-Ca^2+^ were measured in Table 1. The concentrations of PCBMB, BAPTAVP, and Ca^2+^ are the same as 10 μmol/L. In Table 1, the ZP of PCBMB-BAPTAVP (−46.4 mV) is a median between that of PCBMB (−52.1 mV) and BAPTAVP (10.5 mV), which shows that self-assembly may take place between positive charges of BAPTAVP and negative charges of PCBMB. TEM and AFM results further indicate that the self-assembly sphere particles were formed by the coiling from the long polymer chain. Meanwhile, FRET between BAPTAVP and PCBMB greatly increased the fluorescence of BAPTAVP. Adding Ca^2+^, the ZPs of BAPTAVP and PCBMB-BAPTAVP are both enhanced from 10.5 to 27.5 mV for BAPTAVP and from −46.4 to 16.4 mV for PCBMB-BAPTAVP, which shows that BAPTAVP captured Ca^2+^ to effectively increase the fluorescence (23.2 GM for the Φ*δ* of BAPTAVP-Ca^2+^, 169 GM for the Φ*δ* of PCBMB-BAPTAVP-Ca^2+^). In addition, the ^1^HNMR spectra of BAPTAVP, BAPTAVP-added acetylcholine esterase, BAPTAVP-added acetylcholine esterase, and Ca^2+^were measured and are shown in Appendix A. We can see that the carboxyl groups appear in the presence of acetylcholine esterase because of hydrolysis of ester bonds in BAPTAVP (Appendix A). Furthermore, the carboxyl groups will disappear after the addition of Ca^2+^ because of coordination with BAPTAVP, which enhances the fluorescence of BAPTAVP by attenuating the intramolecular charge transfer process [32]. Based on the above results, the possible interaction mechanism is described in Scheme 1.

### 3.7. Fluorescence Microscopy Imaging

The potential application of the PCBMB-BAPTAVP system for the detection of Ca^2+^ in living cells by TPM was investigated (shown in Figure 5). SiHa cells were incubated with 5 µmol/L of BAPTAVPorPCBMB-BAPTAVPfor1h at room temperature and then washed with PBS three times. The fluorescence of BAPTAVP was collected in the range of 520–620 nm (λ_ex_ = 800 nm) with a power of 6%, whereas the fluorescence of PCBMB-BAPTAVP was collected in the range of 450−520 nm at 800 nm with a power of 2.5%, which can avoid the disturbance from the fluorescence of BAPTAVP. Obvious fluorescence from the intracellular area was found when the cells were incubated with the PCBMB-BAPTAVP system at a lower power (shown in Figure 5c), which is different from the poor fluorescence of BAPTAVP (shown in Figure 5a). Furthermore, after the cells were supplemented with 5µmol/L of Ca^2+^ under the same condition, obvious fluorescence of both BAPTAVP and PCBMB-BAPTAVP was found (shown in Figure 5b,d). However, a significantly enhanced fluorescence of the PCBMB-BAPTAVP system from the intracellular area was observed (shown in Figure 5d). The above results indicate that the PCBMB-BAPTAVP system could detect endogenous Ca^2+^ in living cells and was more sensitive. 

## 4. Conclusions

In conclusion, a novel probe with BAPTA as the chelating group and a CP of carbazole derivative were synthesized and characterized. The data including TEM, AFM, DLS, ZPs, ^1^H NMR, and fluorescence spectra showed that the nanoparticles were formed by the self-assembly between positive charges of BAPTAVP and negative charges of PCBMB, which effectively increased the fluorescent properties of BAPTAVP (from 0.098 to 0.39 for Φ; from 16.2 to 89 GM for Φ*δ*) by fluorescence resonance energy transfer (FRET) with PCBMB and attenuating the intramolecular charge transfer (ICT) after the addition of Ca^2+^. The Φ*δ* of PCBMB-BAPTAVP- Ca^2+^ was up to 169 GM. Furthermore, the excellent fluorescence properties of PCBMB-BAPTAVP to detect intracellular Ca^2+^ were demonstrated by TPM. This method may provide a new way to design the system of metal ion detection with better fluorescence.

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
