# Peer review of "Enhanced Two-Photon Fluorescence and Fluorescence Imaging of Novel Probe for Calcium Ion by Self-Assembly with Conjugated Polymer"

_polymers, 2019, doi:10.3390/polym11101643_

Round 1
Reviewer 1 Report
The manuscript by Zhang et al. reported a fluorescence probe for Ca2+ ion based on self-assembly of well-known Ca2+ fluorescence probe system with a conducting polymer. The systems were characterized by TEM, DLS, ZP and spectroscopic measurements. In addition The two photon fluorescence imaging experiments indicate that the coupled system can detect endogeneous Ca2+ in living cell with high sensitivity.
While the approach look interesting the manuscript needs several modifications and clarifications before it could be considered for publication in polymers.
Introduction couldn't sufficiently establish the novelty of the present work or strategy. Why is use of the conducting polymer necessary? What hypothesis worked behind synthesizing that particular polymer. As prerequisite for efficient FRET the polymer requires to exhibit specific emission property (emission maxima etc). Then how can such approach be generalized?
Page-7, line 212: How the concentration is associated with the stoichiometry?
Page-7: line-244: What is the origin of the new absorption band? Is it a charge transfer band? It need to be characterized properly.
Page-9, line-296: It is not clear how the fluorescence peak shifts were correlated with the concentration. Please clarify.
The mechanism of fluorescence enhancement is very crucial, and is not clearly established in the manuscript. The TEM pictures clearly lack any clarity (not sure if magnified version could explain better). Authors can't just rely on microscopy and particle size measurements to establish the self-assembly phenomenon. Did author try to perform other experiment like NMR? It might not be only the ionic interaction and NMR can reveal that.
Fig. 2c need to be improved as the polymer spectra is very unclear.
Fig. 1 graphs need to be labelled appropriately, as no color codes are shown in the current version.
Author Response
Thank you for your confirmation and suggestions, which will help us to improve the quality of our manuscript.
Introduction couldn't sufficiently establish the novelty of the present work or strategy. Why is use of the conducting polymer necessary? What hypothesis worked behind synthesizing that particular polymer. As prerequisite for efficient FRET the polymer requires to exhibit specific emission property (emission maxima etc). Then how can such approach be generalized?
Response: In line 51-77, the introduction has been reorganized to prove the novelty of the present work and describe the idea of design.
Page-7, line 212: How the concentration is associated with the stoichiometry?
Response: The concentration has some contact with the stoichiometry. But for OP and TP excitation, it is not a one-to-one correspondence between the concentration and stoichiometry. The relative explanation has been described in line198-201 on page 6 and the relative reference was supplemented in line 215-216 on page 7.
Page-7: line-244: What is the origin of the new absorption band? Is it a charge transfer band? It need to be characterized properly.
Response: The new absorption peak can be attributed to the results of intermolecular charge transfer. But this is one of reasons. The self-assembly and Fluorescence Resonance Energy Transfer (FRET) play important roles, too. ICT was often verified by the theoretical calculation. So we cited the reported references and analyzed the change of spectra to characterize. The relative description has been supplemented in this paper(in line 259-273 on page 8 ).
Page-9, line-296: It is not clear how the fluorescence peak shifts were correlated with the concentration. Please clarify.
Response: The relation between fluorescence peak shifts and the concentrations has been investigated by supplementing the TP fluorescence spectra of PCBMB-BAPTAVP with the different concentrations in Fig. S1. The relative description has been supplemented in line 318-323 on page 10.
The mechanism of fluorescence enhancement is very crucial, and is not clearly established in the manuscript. The TEM pictures clearly lack any clarity (not sure if magnified version could explain better). Authors can't just rely on microscopy and particle size measurements to establish the self-assembly phenomenon. Did author try to perform other experiment like NMR? It might not be only the ionic interaction and NMR can reveal that.
Response: In order to see clearly the nanoparticles and explore the mechanism, AFM images of BAPTAVP (g), PCBMB-BAPTAVP (h) and PCBMB-BAPTAVP-Ca2+ (L) were supplemented. Moreover, the 1H NMR spectra of BAPTAVP, BAPTAVP + acetylcholin esterase and BAPTAVP + acetylcholin esterase + Ca2+ were measured and showed in supporting information. The relative discussions were supplemented in line 351-353, 364-370 on page 11 and 12. The mechanism diagram has been sketched in Scheme 1.
Fig. 2c need to be improved as the polymer spectra is very unclear.
Response: Fig. 2c has been recreated to replace the old one.
Fig. 1 graphs need to be labelled appropriately, as no color codes are shown in the current version.
Response: Fig. 1 graphs have been re-constructed. The numbers (1-10) have been deleted and the different type lines were used to distinguish.

Reviewer 2 Report
The manuscript is devoted to the study of luminescent response of mixed PCBMB-BAPTAVP system to Ca ions in aqueous and cellular medium (PCBMB – conjugated polymer based on N-bromoethyl-carbazole and 2,5-dimethoxy-1,4-phenylene derivatives; BAPTAVP - tetraethyl 1,2-bis(2-aminophenoxy)-ethane- N,N,N',N'-tetra acetate). Two-photon (TP) excitation technique was used to improve efficiency of luminescent response in living systems. Although sensitization of PCBMB-BAPTAVP luminescence by Ca ions is not very efficient as compared to other studies, this work is a good example of a probe design for TP fluorescence imaging in living cells. I would recommend the manuscript for publication in the Journal after revision of some points.
1. P. 5, lines 128-129: it is unclear how yellow powder of PCBMB was isolated. Did it precipitate from dichloromethane solution upon addition of methyl alcohol?
2. Figs. 1a,b,d and 2d: In my opinion, the numbers 1–10 (for the number of an experiment) in the insets are redundant and overcrowd the graphs. Thus, they can be removed.
3. Fig. 1b, inset: Is Y-axis (F-F0) the difference between intensity of free BAPTAVP and that of BAPTAVP with Ca ion? If so, why the first point (which corresponds to 0 mmol/L of Ca) has the value F-F0 of 5 a.u? Please clarify this.
4. Absorption and emission intensity behavior in Figs. 1a,b (inset) is quite strange: it quickly rises with an increase of concentration of Ca from 0 to 100 mmol/L then unexpectedly gradually decreases with an increase of the concentration from 100 to 2000 mmol/L. Could the Authors explain this phenomenon? Can this be due to further coordination of Ca ions to BAPTAVP to form a 2:1 or 3:1 complex?
5. Emission intensity behavior in Fig. 1d (inset) is even stranger: it decreases then increases then decreases with an increase of concentration of Ca. Is the first point in the graph correct?
6. P. 6, lines 204-205: the sentence “The limit of detection (LOD) of BAPTAVP for OP fluorescence was estimated to be 9.6×10-8 mol/L [25]” is not correct because in Ref [25], another BAPTA derivative was used.
7. In my opinion, for clarity, X-axis in Fig. 2d (inset) and legend in Fig. 2c should be responsible for PCBMB/BAPTAVP molar ratio (as in Figs. 1a,b,d), but not for volume of PCBMB.
8. Figs. 2b,d: What are satellite sharp bands at ca. 520 nm? Why are they absent in Figs. 3a,b? Can they be due to the presence of the second harmonics of a beam source? If so, why intensity of these bands increases with increase of the main bands?
9. What are left, central and right images in Figs. 5a-d? Please clarify shooting conditions in the caption.
Author Response
Thank you for your care, valuable comments and suggestions, which will be benefit to the improving quality of the manuscript and our research. The detailed corrections are listed point by point as following.
1.P. 5, lines 128-129: it is unclear how yellow powder of PCBMB was isolated. Did it precipitate from dichloromethane solution upon addition of methyl alcohol?
Response: The purified process of PCBMB was re-described in line 150 on page 5 in orderto help us understand it more clearly.
Figs. 1a,b,d and 2d: In my opinion, the numbers 1–10 (for the number of an experiment) in the insets are redundant and overcrowd the graphs. Thus, they can be removed.
Response: Fig.1a,b,d have been removed the numbers and re-constructed.
Fig. 1b, inset: Is Y-axis (F-F0) the difference between intensity of free BAPTAVP and that of BAPTAVP with Ca ion? If so, why the first point (which corresponds to 0 mmol/L of Ca) has the value F-F0 of 5 a.u? Please clarify this.
Response: We are so sorry for our mistake. It has been corrected in the inset of Fig.1b.
Absorption and emission intensity behavior in Figs. 1a,b (inset) is quite strange: it quickly rises with anq increase of concentration of Ca from 0 to 100 mmol/L then unexpectedly gradually decreases with an increase of the concentration from 100 to 2000 mmol/L. Could the Authors explain this phenomenon? Can this be due to further coordination of Ca ions to BAPTAVP to form a 2:1 or 3:1 complex?
Response: The fluorescence change of probe was related to the concentrations of Ca2+, which may be attributed to the structure of coordination complex. The relative state and literature were supplied in line 198-201 on page 6.
Emission intensity behavior in Fig. 1d (inset) is even stranger: it decreases then increases then decreases with an increase of concentration of Ca. Is the first point in the graph correct?
Response: The first point in Fig. 1d has corrected and the Fig. 1d has been re-sketched. Some relative state was supplemented in line 198-201 on page 6 and line215-216 on page 7.
P. 6, lines 204-205: the sentence “The limit of detection (LOD) of BAPTAVP for OP fluorescence was estimated to be 9.6×10-8 mol/L [25]” is not correct because in Ref [25], another BAPTA derivative was used.
Response: We are sorry for the inappropriate description to cause your misunderstanding. The limit of detection is calculated according to the reported method, and the value was compared with that in Ref [25]. The detailed description has been supplemented in line 204-207 on page 6.
In my opinion, for clarity, X-axis in Fig. 2d (inset) and legend in Fig. 2c should be responsible for PCBMB/BAPTAVP molar ratio (as in Figs. 1a,b,d), but not for volume of PCBMB.
Response: Fig.1 (a, b, d) and Fig. 2d have been re-sketched. The molar ratio replaced the volume.
Figs. 2b,d: What are satellite sharp bands at ca. 520 nm? Why are they absent in Figs. 3a,b? Can they be due to the presence of the second harmonics of a beam source? If so, why intensity of these bands increases with increase of the main bands?
Response: Yes, you are right. The sharp bands at 528 nm are mainly the second harmonics of a beam source (264 nm). Their amplitudes are enhanced with the increased concentrations of PCBMB, the relative reference was supplemented in line 259-263 on page 8. It wasn’t found in Fig.3a because the concentration of polymer was lower (line 4 in Fig.2 d) with a non-obvious sharp band. The excitation wavelength was at 800 nm in Fig.3b, so it isn’t possible to see the second harmonics of the beam source.
What are left, central and right images in Figs. 5a-d? Please clarify shooting conditions in the caption.
Response: The left, central and right images in Figs. 5a-d have been clarified in the photos and caption.
Round 2
Reviewer 1 Report
I'm mostly satisfied with the revision made by the authors and would recommend the acceptance after minor modification as stated below.
Line 269-272: It is not clear to me what do author mean by "...shortening distance because of overlap between fluorescence emission spectrum and UV absorption spectrum..."
Please clarify or modify the sentence.
Author Response
Thank you for your further comments. The detailed corrections are listed as following.
1. Line 269-272: It is not clear to me what do author mean by "...shortening distance because of overlap between fluorescence emission spectrum and UV absorption spectrum..." Please clarify or modify the sentence.
Response: The sentences have been reorganized for a better understanding. The fonts in line 270-273 on page 8 has been selected by clicking on the bold button.
